# YCl_3_-Substituted CsPbI_3_ Perovskite Nanorods for Efficient Red-Light-Emitting Diodes

**DOI:** 10.3390/nano13081366

**Published:** 2023-04-14

**Authors:** Muhammad Imran Saleem, Amarja Katware, Al Amin, Seo-Hee Jung, Jeong-Hwan Lee

**Affiliations:** 13D Convergence Center, Inha University, Incheon 22212, Republic of Korea; 2Department of Materials Science and Engineering, Inha University, Incheon 22212, Republic of Korea

**Keywords:** anisotropic growth, YCl_3_-substituted perovskite, 1D nanorods, transition dipole moments, outcoupling efficiency

## Abstract

Cesium lead iodide (CsPbI_3_) perovskite nanocrystals (NCs) are a promising material for red-light-emitting diodes (LEDs) due to their excellent color purity and high luminous efficiency. However, small-sized CsPbI_3_ colloidal NCs, such as nanocubes, used in LEDs suffer from confinement effects, negatively impacting their photoluminescence quantum yield (PLQY) and overall efficiency. Here, we introduced YCl_3_ into the CsPbI_3_ perovskite, which formed anisotropic, one-dimensional (1D) nanorods. This was achieved by taking advantage of the difference in bond energies among iodide and chloride ions, which caused YCl_3_ to promote the anisotropic growth of CsPbI_3_ NCs. The addition of YCl_3_ significantly improved the PLQY by passivating nonradiative recombination rates. The resulting YCl_3_-substituted CsPbI_3_ nanorods were applied to the emissive layer in LEDs, and we achieved an external quantum efficiency of ~3.16%, which is 1.86-fold higher than the pristine CsPbI_3_ NCs (1.69%) based LED. Notably, the ratio of horizontal transition dipole moments (TDMs) in the anisotropic YCl_3_:CsPbI_3_ nanorods was found to be 75%, which is higher than the isotropically-oriented TDMs in CsPbI_3_ nanocrystals (67%). This increased the TDM ratio and led to higher light outcoupling efficiency in nanorod-based LEDs. Overall, the results suggest that YCl_3_-substituted CsPbI_3_ nanorods could be promising for achieving high-performance perovskite LEDs.

## 1. Introduction

Metal halide perovskites (MHPs) have attracted considerable attention due to their impressive characteristics, such as exceptional color purity, effective radiative recombination, adjustable emission wavelengths, and inexpensive solution processability, making them a promising contender for the next generation of lighting and display [1,2,3,4,5]. All inorganic CsPbI_3_ nanocrystals (NCs) are indispensable for these purposes among various MHPs due to their exceptional thermal and chemical durability [5]. Recent strategies, including morphology and interfacial management, the encapsulation of perovskite nanocrystals within polymers or glasses, architecture engineering, and surface chemistry engineering, have been employed to enhance the practical applications of CsPbI_3_ NCs and raise the external quantum efficiency (EQE) of CsPbI_3_ NC-based light-emitting diodes (LEDs) [3,4,5,6]. Despite the significant advancements, CsPbI_3_ NC LEDs have considerable limitations due to their small size (0-dimensional NCs), which is caused by the confinement effect [7]. Due to their small size, the defect-enriched surface of colloidal CsPbI_3_ NCs has a detrimental effect on the photoluminescence quantum yield (PLQY) and the low-luminous efficiency of LEDs [8]. Furthermore, the excessive native ligands required to passivate the large surface area of small-sized CsPbI_3_ NCs forms an insulation layer that impedes the effective carrier transport capability in the assembled NCs’ film, resulting in numerous challenging issues associated with CsPbI_3_ NC-based LEDs [2].

The limitations of conventional CsPbI_3_ NC-based LEDs can be overcome by using anisotropic one-dimensional (1D) nanorods made from CsPbI_3_ NCs. This is due to the combination of two-directional quantum confinement effects and a significantly reduced surface trap density in 1D nanorods, leading to excellent photophysical properties and high aspect ratios [9]. The unique surface morphology of 1D nanorods contributes to their excellent photophysical properties and high aspect ratios [10,11], and their well-defined morphology can restrict the active region of charge carriers and reduce the carrier transport distance [3], making them more conducive to the development of efficient LEDs compared to their counterpart nanocrystals [12].

Although the synthesis of perovskite nanorods from the water–oil transformation of Cs_4_PbBr_6_ polyhedrons into nanorods or fragmentation of perovskite nanowires initiated by anion-exchange processes has been reported recently [13,14], these methods produce impure morphologies and defect-enriched surfaces. Hence, it is still challenging to directly synthesize perovskite nanorods with high PLQY and well-defined aspect ratios that exhibit efficient radiative recombination rates. Importantly, the anisotropic nature of a nanorod-based emissive layer is valuable for further increasing the outcoupling efficiency by overcoming the photon losses that become trapped through waveguiding and total internal reflection, as they have high ratios of horizontal transition dipole moments (TDMs) compared to isotopically-oriented nanocubes (nanocrystals) [15,16,17].

Here, we propose a proper method to directly synthesize perovskite nanorods composed of YCl_3_-substituted CsPbI_3_ by using Yttrium (III) Chloride Hexahydrate (YCI_3_·6H_2_O). This approach reduces the crystal size of CsPbI_3_ NCs by partially substituting Pb^2+^ and I^−^ ions with Y^3+^ and Cl^−^ ions. The YCl_3_ passivates surface traps and controls the net recombination rates, significantly improving PLQY. Moreover, the environmental durability of the YCl_3_-substituted CsPbI_3_ nanorods is significantly enhanced, with only a 28% decrease in PLQY after 45 days of storage under ambient conditions. Importantly, the YCl_3_:CsPbI_3_ nanorod-based LEDs exhibit a peak EQE of 3.16%, 1.86 times higher than that of the control CsPbI_3_ NC-based device (1.69%). Furthermore, the ratio of horizontal transition dipole moments of the anisotropic YCl_3_:CsPbI_3_ nanorods is 75%, higher than that of isotopically-oriented TDMs in CsPbI_3_ nanocrystals (67%), resulting in higher light outcoupling efficiency. Thus, our findings suggest that anisotropic nanorods have promising potential in light-emitting devices.

## 2. Materials and Methods

All the chemicals were purchased from Sigma-Aldrich.

### 2.1. Synthesis of Unsubstituted and YCl_3_-Substituted CsPbI_3_ NCs

The previously reported hot injection method was employed to synthesize the unsubstituted and YCl_3_-substituted CsPbI_3_ NCs [18,19].

#### 2.1.1. Cesium Oleate Preparation

Firstly, Cs_2_CO_3_ (2.49 mmol), OLA (2.5 mL), and 30 mL ODE were degassed and dried at 120 °C for 1 h. Subsequently, the temperature was raised to 150 °C until a clear solution was obtained under N_2_ flow.

#### 2.1.2. Synthesis of CsPbI_3_ NCs

PbI_2_ (0.174 g, 0.376 mmol) and ODE (10 mL) were loaded into a 100 mL three-neck flask, degassed, and dried under vacuum for 1 h at 120 °C. Then, OA (1 mL) and OLA (1 mL) preheated to 70 °C were injected under the protection of N_2_. After the solution became clear, the temperature was raised to 170 °C, and 0.8~1 mL Cs-oleate (pre-heated to >100 °C) was swiftly injected, which was quenched by immersing the flask in an ice-water bath (5 s later).

#### 2.1.3. Synthesis of YCl_3_:CsPbI_3_ NRs

ODE (10 mL), OA (1 mL), and YCl_3_·6H_2_O (0.184 mmol) were loaded into a three-neck flask and adequately dissolved at 150 °C for 1 h under N_2_ flow. Then, PbI_2_ (0.174 g, 0.376 mmol) and OLA were added, and the temperature was raised to 170 °C (30 min later). Then, the prepared Cs-oleate was quickly injected into the mixture and 60~65 s later, the reaction was quenched by immersing the flask in an ice-water bath [12].

#### 2.1.4. Purification

The unsubstituted and YCl_3_-substituted CsPbI_3_ NCs were separated by centrifugation at 5000 rpm for 10 min to remove the ODE and unreacted ligands. The precipitate was dispersed in hexane/toluene, and then anti-solvent methyl acetate was mixed, followed by centrifugation at 10,000 rpm for 10 min. The mixture was dispersed in 10 mL hexane/toluene and stored in the refrigerator. After 24 h, the supernatant required colloidal ink for LED application.

### 2.2. LED Fabrication

150 nm thick indium tin oxide (ITO) patterned glass substrates were cleaned by sequential ultrasonication in acetone and isopropanol for 15 min each. Subsequently, the UV-ozone treatment was carried out for 15 min to improve hydrophilicity before drying with N_2_ flow. PEDOT:PSS (filtered with 0.45 µm PVDF filter) was then spin-coated for 1 min at 4000 rpm, which was then annealed at 140 °C for 30 min. All the substrates were transferred into the N_2_-filled glove box. The hole-transporting Poly-TPD (4 mg/mL in CB) layer was spin-coating onto PEDOT:PSS films at 4000 rpm and baked at 120 °C for 15~20 min. The perovskite emissive layer was spin-coated at a speed of 2000 rpm for 45 s. Then, 1,3,5-tris(1-phenyl-1H-benzimidazol-2-yl)benzene (TPBi, 40 nm), LiF (1 nm), and Al (130 nm) were sequentially deposited by thermal evaporation into a vacuum deposition chamber. All the devices were encapsulated by a glass lid with a UV-curable resin in an N_2_-filled glovebox. The optoelectronic properties of LEDs were analyzed using a semiconductor parameter analyzer (Keithley 237) connected with a spectrophotometer (Photo Research PR-670). The UV-Vis-IR absorption spectrum of the unsubstituted and YCl_3_-substituted CsPbI_3_ NC film was tested using a PerkinElmer LAMBDA-900 spectrophotometer. PLQY was measured by Quantaurus-QY Absolute PL quantum yield spectrometer (Hamamatsu, C11347-11). The structure of the unsubstituted and YCl_3_-substituted CsPbI_3_ NCs’ thin film was analyzed by an X-ray diffractometer (X’Pert-PRO MRD, Phillips). The shape of the unsubstituted and YCl_3_-substituted CsPbI_3_ NCs was confirmed using a field-emission transmission electron microscope (FE-TEM, JEM-2100F) and the cross-sectional TEM images of the LED were obtained from a Cs-corrected TEM (JEM-ARM 200F, JEOL) installed in the Center for University-wide Research Facilities (CURF) at Jeonbuk National University. Chemical analysis was conducted by X-ray Photoelectron Spectrometer (XPS).

## 3. Results and Discussion

The control (unsubstituted) and YCl_3_-substituted CsPbI_3_ NCs were synthesized following the two-step hot injection method [20]. The YCl_3_-doped CsPbI_3_ NRs were realized by adding 0.184 mmol of YCl_3_·6H_2_O into perovskite medium followed by the injection of Cs-oleate precursor (Figure 1a). This is unlike the previous reports, which aimed at metal chlorides that used either the identical or adjoining chloride ion in the pristine perovskite nanocrystals [14,21]. This study examines how metal chloride can modulate the shape and optoelectronic properties of CsPbI_3_ NCs. Besides changing the surface defect of perovskite NCs, the YCl_3_-doping has an additional effect on inducing the anisotropic growth of the crystals. The intention doping of YCl_3_ was carried out in the CsPbI_3_ NCs’ reaction medium. The distinctive chloride (Cl^−^) was not proximate with the iodide (I^−^) ions of the CsPbI_3_ NCs. The presence of chloride ions on the surface of perovskite nanocrystals (NCs) and the varied bond energies among chloride (Cl^−^) ions and iodide (I^−^) ions are responsible for the anisotropic growth of perovskite NCs. (Figure 1b). The transmission electron microscopy (TEM) analyses provide evidence to support this conjecture, as shown in Figure 1. The TEM and high resolution (HR-TEM) morphology of the control and YCl_3_-substituted CsPbI_3_ NCs is revealed by TEM (Figure 1). The control CsPbI_3_ NCs contain monodisperse and regular cubic shapes (Figure 1a). The HR-TEM images show a high crystallinity and lattice spacing of 6.2 Å of the CsPbI_3_ NCs, corresponding to the (100) plane of cubic perovskite (Figure 1b,c) [22,23]. The average particle size of the control CsPbI_3_ NCs is determined to be ~10.05 nm (Figure 1d). Remarkably, the 0.184 mmol YCl_3_-substituted CsPbI_3_ NCs show the one-dimensional nanorods (NRs) (Figure 1e–g). The HR-TEM image of YCl_3_-substituted CsPbI_3_ NRs displays a lattice spacing of 4.5 Å, corresponding to the (110) plane of perovskite (Figure 1g) [12]. The aspect ratio of the YCl_3_-substituted CsPbI_3_ NCs is ~2.3, and the average sizes of length and diameter are 18.5 and 8.2 nm, respectively.

The X-ray diffraction (XRD) patterns were conducted to ascertain the crystal structure of CsPbI_3_ NCs and YCl_3_:CsPbI_3_ NRs, as shown in Figure 2. Both the CsPbI_3_ NCs and YCl_3_:CsPbI_3_ NRs adhere to the reference pattern of the bulk cubic CsPbI_3_ perovskite (PDF#98-018-1288), and the diffraction peaks, which appear at 14.02°, 20.03°, 28.407°, 31.88°, 35.45°, 41.05°, and 51.63°, are corresponding to cubic planes of (100), (110), (200), (210), (211), (220), and (300), respectively. As shown in Figure 2b,c, the angles of diffraction peaks for the (100) and (200) planes shifted towards higher values attributed to the reduction in the lattice parameters of YCl_3_:CsPbI_3_ NRs, which stemmed from the partial substitution lead cation (Pb^2+^) and iodide ions (I^−^) with Y^3+^ cation and Cl^−^ ions, respectively [12,24]. The scanning electron microscopy (SEM) images of the unsubstituted and YCl_3_-substituted CsPbI_3_ NCs are illustrated in Figure 2d,e. Contrary to perovskite NCs, the SEM image of (YCl_3_-substituted CsPbI_3_) nanorods are homogeneously distributed on the glass substrate, which indicates that the nanorod film layer has a good foundation for electroluminescence devices.

X-ray photoelectron spectra (XPS) analysis was performed to gain insight into the interaction of YCl_3_ with CsPbI_3_ nanocrystals (NCs), and the results are presented in Figure 3a. The characteristic XPS signals for Cs 3d, Pb 4f, I 3d, Y 3d, and Cl 2p were observed in YCl_3_-doped CsPbI_3_ NCs and N 1s, O 1s, and C 1s signals coupled with native ligand bonding. The high-resolution XPS spectra of Cs 3d, Pb 4f, I 3d, Y 3d, and Cl 2p are displayed in the order in Figure 3b–f. The partial substitution of I^−^ ions by Cl^−^ ions is evidenced by an increase in the binding energies of Pb^2+^ 4f_5/2_ and Pb^2+^ 4f_7/2_ from 143.56 and 138.37 eV to 143.64 and 138.7 eV, respectively. The binding energy of I 3d and Cs 3d displays little variation compared to YCl_3_:CsPbI_3_ nanocrystals (NCs). More importantly, the binding energy signals of Y^3+^ and Cl^−^ can be observed in the YCl_3_-substituted CsPbI_3_ nanorods (NRs). These findings provide additional evidence supporting the partial substitution of iodide ions by chloride ions [25].

To gain a better understanding of the effects of the partial substitution of Pb^2+^ cation and iodide ions (I^−^) with yttrium cation (Y^3+^) and chloride (Cl^−^) ions, the optical properties of both as-synthesized CsPbI_3_ nanocrystals (NCs) and YCl_3_:CsPbI_3_ nanorods (NRs) were analyzed (Figure 4). The corresponding normalized PL and absorption spectra of YCl_3_:CsPbI_3_ NRs showed that the peak position exhibited a blue shift owing to the introduction of chloride (Cl^−^) ions (Figure 4a,b) [26]. The enlarged bandgap caused the blue shifts of the absorption and PL spectra for YCl_3_:CsPbI_3_ NRs due to the partial replacement of lead (Pb^2+^) cation and iodide (I^−^) ions with yttrium cation (Y^3+^) and chloride (Cl^−^) ions [27]. The blue dots in Figure 4b demonstrate the blue shift associated with introducing YCl_3_ into the CsPbI_3_ perovskite. The Tauc plot of the unsubstituted and YCl_3_-substituted CsPbI_3_ NC films is illustrated in Figure 4c,d, and the corresponding band value of the unsubstituted is 1.76 eV, and the YCl_3_-substituted is 1.82 eV. The PLQY increased from 51% to 70% for the YCl_3_ passivated CsPbI_3_ NCs, suggesting enhanced radiative recombination followed by yttrium chloride doping (Figure 4e). The environmental durability of the YCl_3_:CsPbI_3_ solution was noticeably enhanced (Figure 4f), and the YCl_3_-substituted CsPbI_3_ NRs maintained (50% out of 70%) a PLQY, with a loss of 28% PLQY after being stored for 45 days under ambient conditions, prized for the effectiveness of yttrium chloride passivation. However, over the same period of time, the PL quantum yield of pristine CsPbI_3_ NCs nearly approached zero. The inset of Figure 4f shows the images recorded at different times for the unsubstituted and YCl_3_-substituted CsPbI_3_ NC solution.

The control and YCl_3_-substituted CsPbI_3_ NCs were employed as emitters to evaluate the potential applications in perovskite LEDs. The LEDs were fabricated based on the configuration of indium tin oxide (ITO)/poly(3,4-ethylenedioxythiophene) polystyrenesulfonate (PEDOT:PSS)/poly(4-butylphenyl-diphenyl-amine) (P-TPD)/unsubstituted CsPbI_3_ or YCl_3_:CsPbI_3_/1,3,5-tris(1-phenyl-1H-benzimidazol-2-yl)benzene (TPBi)/lithium fluoride (LiF)/aluminum (Al). The schematic illustration of PeLED and the corresponding energy levels diagram of the functional layer are illustrated in Figure 5a,b. The functional layers’ energy level values are taken from previous literature [12]. The thickness of ITO (150 nm), PEDOT: PSS, P-TPD (60 nm), YCl_3_:CsPbI_3_ (55 nm), TPBi (60 nm), and LiF/Al (130 nm) were analyzed by the cross-sectional TEM image, as shown in Figure 5c. The current density-voltage-luminance (*J-V-L*) curves of the unsubstituted and YCl_3_-substituted CsPbI_3_ NC LEDs with a 4 mm^2^ emitting area are displayed in Figure 5d. The turn-on voltage (where the luminance achieved 1 cd/m^2^) is reduced from ~3.9 V to ~3.6 V for the YCl_3_-substituted CsPbI_3_ NC LED, revealing that more balanced carriers injected owing to their matched energy level with the carrier transfer layer and enhanced conductivity induced by the YCl_3_-substitution that facilitated the efficient charges’ injection [12,23]. The CsPbI_3_ NC and YCl_3_:CsPbI_3_ NR-based LEDs showed a maximum luminance of 263.1 cd/m^2^ and 421.8 cd/m^2^, respectively. The electroluminescence (EL) spectra of the unsubstituted and YCl_3_-substituted CsPbI_3_ NC LEDs were observed at 691 nm and 688 nm (Figure 5e). The values are quite different from the PL spectra in the case of the unsubstituted one (Figure 4a). The similar EL spectra of the two devices are attributed to the weak microcavity effect in the LED structure, which is related to the recombination zone of the device. The recombination zone would be at the interface of EML and ETL in both cases due to the dominance of the hole carrier in perovskite EMLs. The YCl_3_:CsPbI_3_-based LEDs revealed high color purity with Commission internationale de l’éclairage (CIE) coordinates of (0.71, 0.26), corresponding to the BT. 2020 color gamut, as shown in Figure 5f. The EQE vs. luminance curves are displayed in Figure 5g. The peak EQE of YCl_3_:CsPbI_3_ is 3.16%, 1.86-fold higher than the pristine CsPbI_3_ NC (1.69%) based LED. This enhancement is attributed to enhanced PLQY and more balanced carrier transfer in the YCl_3_:CsPbI_3_ EML layer.

To further investigate the enhanced EQE in the YCl_3_-substituted CsPbI_3_ NRs, we conducted angle-dependent photoluminescence (ADPL) measurements to probe the orientation of transition dipole moments (TDMs) in assembled thin film of CsPbI_3_ nanocrystals/nanorods. The outcoupling efficiency in PeLEDs has the potential for improvement through controlling the orientation of TDMs [28,29,30]. The optical TDMs of nanoplatelets and nanorods are highly anisotropic, and outcoupling efficiency in planer PeLEDs is profoundly associated with the orientation of emissive TDMs [15,17]. The orientation of the optical TDMs of the unsubstituted and YCl_3_-substituted CsPbI_3_ NCs were measured by the ratio of horizontal TDMs (Θ). The experimental data are fitted to the pattern simulated, employing the classical dipole radiation model [31]. The Θ values of CsPbI_3_ NCs’ and YCl_3_:CsPbI_3_ NRs’ films are determined to be 67% and 75% (Figure 6a,b). The Θ value in anisotropic nanorods is considerably higher than that in isotropic nanocrystals. Thus, the optical TDMs that are horizontally oriented in anisotropic nanorods are preferred for light outcoupling, resulting in the improved EQE of the LEDs.

## 4. Conclusions

In this study, we investigated the effects of incorporating metal chloride (YCl_3_) into CsPbI_3_ nanocrystals to control the dimensions. We found that the incorporation of YCl_3_ led to a decrease in the lattice parameters of the CsPbI_3_ nanocrystals, which resulted from the partial substitution of larger lead cation (Pb^2+^) and iodide (I^−^) ions with smaller Y^3+^ and Cl^−^ ions. The presence of Cl^−^ ions on the surface of the NCs, coupled with the difference in bond energies between chloride (Cl^−^) and iodide (I^−^) ions, led to the anisotropic formation of the CsPbI_3_ nanocrystals into one-dimensional (1D) nanorods. The YCl_3_ also significantly improved the photoluminescence quantum yield and storage lifetime of the perovskite solution by passivating nonradiative recombination rates and defects properly. Finally, we used the YCl_3_-substituted CsPbI_3_ nanorods as the emissive layer in red LEDs and observed a significant improvement in their performance. The LEDs exhibited an external quantum efficiency of 3.16% which is 1.86-fold higher than the pristine CsPbI_3_ NC (1.69%) based LED, attributed to the improvement of the ratio of horizontal TDMs in the anisotropic YCl_3_:CsPbI_3_ nanorods to 75% from 67% of that of NCs. Overall, the combined characteristics of YCl_3_-substituted CsPbI_3_ nanorods show great potential for developing stable and efficient red LEDs.

## Data Availability

Data will be made available upon request.

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
