# Peer review of "YCl3-Substituted CsPbI3 Perovskite Nanorods for Efficient Red-Light-Emitting Diodes"

_nanomaterials, 2023, doi:10.3390/nano13081366_

Round 1

Reviewer 1 Report

Reviewer comments:

In this work, the authors report on the enhanced photoluminescence properties of CsPbI3 PNCs upon incorporating YCl3 metal chloride. Both PLQY and stability of the so-formed PNCs were improved when compared to pristine CsPbI3 PNCs, rendering the former material a suitable candidate for LED applications. Based on this, the authors present the enhanced performance of the developed red light LED. The paper is well written, while presenting interesting results for the reader. However, a few points need to be clarified and addressed. Thus, in order to be considered for publication in Nanomaterials (MDPI), the authors are strongly encouraged to address the points listed below (major revisions) that are critical for strengthening the paper.

 Suggested revisions and recommendations:

1. Page 1, lines 33-37: The authors refer to recent strategies for improving the efficiency of PNCs. However, the authors need to highlight parallel strategies for improving stability of PNCs, as well as, lead toxicity issues, i.e. generally acknowledged critical issues of the field.  Following this, the authors are kindly asked to refer to the two provided solutions for resolving these problems, i.e. encapsulation within polymers and glasses (S. N. Raja et al., ACS Appl. Mater. Interfaces 8, 35523, 2016, and I. Konidakis et al., Nanoscale 14, 2966, 2022). These approaches render the application of PNCs plausible for realistic employment in photonic and optoelectronic applications, i.e. including LEDs.

 2. Page 4, Figure 1d, line 165: The authors are kindly asked to change the scale of x-axis from 5 to 15 nm, instead of the employed 0 to 20 nm, for better magnification of the histogram.

 3. Page 6, lines 208-209: The authors state that the obtained blue-shift of the PL feature is due to the partial substitution of Y3+ cation. Do they mean the partial substitution of Pb2+ by the Y3+ cation? Is it not the introduction of Cl- that also contributes to the obtained blue shift?

 4. Page 6, 1st paragraph: Figure 1b is not mentioned in the text. Indeed, do the authors point some shift obtained from the absorbance spectra with the dotted blue lines, or they just point out where the PL was exhibited?

 5. Page 6, lines 214-218: The authors consider the stability of the PLQY over time (Figure 4d). This is a nice part of the study. However, while within the text a period of 45 days is mentioned, the plot only presents up to 40 days. Please modify accordingly. Also, would it be possible to add error bars in Figure 4d?

Reviewer 2 Report

The paper studied the effect of the incorporation of YCl3 into CsPbI3 NCs. The incorporation of YCl3 can change the morphology of CsPbI3 NCs, leading to improved PL and EL efficiency. These results are interesting. I would recommend the publication of the paper after addressing the following questions.

1, About the scheme. I am confused by the scheme 1a. It seems that the CsPbI3 NCs were prepared at 150 C, while CsPbI3 NRs were prepared at 185 C. However, the synthesis temperatures are the same in the experimental section, both of which are 170 C.

2, The anisotropic growth behavior was not well explained. Since CsPbI3 has an isotropic crystal structure, the effect of the Cl- ions should be the same for the exposed surfaces.

3, About the TEM results. It seems that most of the NCs are cubic from figure 1e. The most difference between 1e and 1a is the size distribution of the NCs. The distribution of length/width ratio is more convincing for illustration the formation of NRs if it is not obvious.

4. The figure caption of figure 2 is totally wrong!

5. The shifts of the XRD peaks are not obvious. Magnified part is needed to show the shift.

6. The presence of the shift of the absorption spectra is too arbitrary. It is better to give Tauc plots and estimate the optical band gaps of the materials to show the differences.

7. Why the shift for the PL spectrum is larger than the EL spectrum for the doped one?

8. Do the increased size of the NCs contribute to the improved PL & PL efficiency except the change of the shape?

Round 2

Reviewer 1 Report

The authors have addressed all the comments.